# IgG Fusion Proteins for Brain Delivery of Biologics via Blood–Brain Barrier Receptor-Mediated Transport

**DOI:** 10.3390/pharmaceutics14071476

**Published:** 2022-07-15

**Authors:** Ruben J. Boado

**Affiliations:** Department of Medicine, University of California, Los Angeles (UCLA), Los Angeles, CA 90095, USA; boado@ucla.edu

**Keywords:** blood–brain barrier, protein-based therapy, monoclonal antibody, insulin receptor, transferrin receptor, lysosomal storage disorders, fusion proteins, Parkinson’s disease, Alzheimer’s disease, neurotrophic factors, decoy receptors

## Abstract

The treatment of neurological disorders with large-molecule biotherapeutics requires that the therapeutic drug be transported across the blood–brain barrier (BBB). However, recombinant biotherapeutics, such as neurotrophins, enzymes, decoy receptors, and monoclonal antibodies (MAb), do not cross the BBB. These biotherapeutics can be re-engineered as brain-penetrating bifunctional IgG fusion proteins. These recombinant proteins comprise two domains, the transport domain and the therapeutic domain, respectively. The transport domain is an MAb that acts as a molecular Trojan horse by targeting a BBB-specific endogenous receptor that induces receptor-mediated transcytosis into the brain, such as the human insulin receptor (HIR) or the transferrin receptor (TfR). The therapeutic domain of the IgG fusion protein exerts its pharmacological effect in the brain once across the BBB. A generation of bifunctional IgG fusion proteins has been engineered using genetically engineered MAbs directed to either the BBB HIR or TfR as the transport domain. These IgG fusion proteins were validated in animal models of lysosomal storage disorders; acute brain conditions, such as stroke; or chronic neurodegeneration, such as Parkinson’s disease and Alzheimer’s disease. Human phase I–III clinical trials were also completed for Hurler MPSI and Hunter MPSII using brain-penetrating IgG-iduronidase and -iduronate-2-sulfatase fusion protein, respectively.

## 1. Introduction

The hematoencephalic or blood–brain barrier (BBB) is the anatomical and molecular barrier that separates in vivo the brain from the blood. This barrier evolved to prevent the transport to the brain of peripheral neurotransmitters, cytokines, and microorganisms, which may produce deleterious, if not lethal, effects in the central nervous system (CNS). The characteristics of this barrier have been extensively reviewed, and it is basically only permeable to lipophilic molecules of <400 Da [1,2,3,4]. Thus, histamine, a small polar molecule of 110 Da does not cross the BBB [2]. Hydrophobic nutrients of low molecular weight gain access to the brain through the BBB via facilitated transporters, as in the case of GLUT1 for glucose and LAT1 for large neutral amino acids [5,6]. Proteins, in general, do not cross the BBB; however, there are a few exceptions where proteins produced in peripheral organs gain access to the brain via receptor-mediated transcytosis, as in the case of insulin, transferrin, leptin, and insulin-like growth factor [7,8,9,10]. Targeting these BBB endogenous transporters with monoclonal antibodies gained attention in the early 1990s, and an in vivo demonstration of the efficacy of a brain-penetrating construct was published using vasopressin intestinal peptide (VIP) conjugated to the OX26 monoclonal antibody to the rat transferrin receptor using the avidin–biotin technology [11]. The administration of the OX26-avidin-biotinylated-VIP produced a marked increase in the brain blood flow. On the contrary, the biotinylated-VIP had no effect in the brain, as it does not cross the BBB [11]. The construction and efficacy of chemical conjugates targeting either the transferrin or the insulin receptor in rodents and non-human primates have been reported [12,13,14,15].

With the cloning of monoclonal antibodies to the mouse transferrin (TfRMAb) and human insulin (HIRMAb) BBB receptors, respectively [16,17], the engineering of bifunctional IgG fusion proteins was possible [18,19,20]. These fusion proteins comprise a transport domain, i.e., TfRMAb or HIRMAb, and a therapeutic domain fused to the C-terminus of either the heavy or light chain of the transporting MAb. Thus, any potential protein therapeutic can be transported through the BBB into the brain in the form of a fusion protein targeting a BBB-receptor-mediated transport (Figure 1). In this schematic representation, a protein therapeutic in fused to the C-terminus of the transporting MAb, which binds to a BBB receptor, inducing the transport of the fusion protein through the BBB. The binding of the MAb fusion protein to its BBB receptor does interfere with the binding of its endogenous ligand, so both the ligand and the MAb fusion are transported through the BBB and released into the brain interstitial fluid. Depending on the characteristics of the therapeutic domain, the fusion protein can (i) target a receptor on the surface of brain cells, as in the case of neurotrophic factors; (ii) bind and inactivate a target molecule, as in the case of decoy receptors and bispecific MAbs; and (iii) be internalized in brain cells via receptor-mediated endocytosis through the same transport systems used to cross the BBB, as in the case of enzymes for the treatment of lysosomal storage disorders (LSD) and/or bispecific MAbs. A detailed mathematical model of receptor-mediated transport across the BBB was recently published [21]. A generation of IgG fusion proteins targeting both the human and mouse BBB transport systems has been engineered (Table 1 and Table 2). The aim of this article is to review this generation of IgG fusion proteins.

## 2. Genetic Engineering of IgG Fusion Proteins

The genetic engineering of IgG fusion proteins has been performed using either individual expression vectors for light- and heavy-chain expression genes or tandem vectors carrying both light- and heavy-chain expression genes [22,23,24,25,26,27,28,29,30,31,32,33,34,35,36,37,38,39,40,41,42]. The cDNA corresponding to the mature therapeutic domain (without the signal peptide) is ligated into the C-terminus of the appropriate expression gene via designed restriction endonuclease sites, which provides a short linker composed of 2–4 serine residues. The therapeutic gene can be inserted in either the heavy or light chain of the transport MAb, and a few examples are shown in Figure 2. In particular cases, the short linker approach produces suboptimal levels of enzyme activity and/or production, which may be restored by the introduction of a long 31-amino-acid linker corresponding to the IgG3 hinge region [27,28]. The engineering of IgG fusion proteins may be performed by fusing the therapeutic domain on the N-terminus of the transport MAb, i.e., the heavy chain of the MAb. However, it was demonstrated using a glucuronidase (GUSB) fusion protein that this construct had a marked reduction in the affinity for the target receptor, to levels that would abolish its transport through the BBB [43]. For studies in rodents, the engineering of IgG fusion proteins has been performed targeting the mouse or rat transferrin receptor [17,19,20,44]. For studies in humans and non-human primates, the IgG fusion proteins were initially produced with an MAbs directed to the human insulin receptor, and lately the MAbs have been directed to the human transferrin receptor as well [16,24]. The anti-human insulin receptor MAb cross-reacts with the BBB insulin receptor of old-world primates, such as the rhesus monkey [45]. Targeting the rodent transferrin receptor or the human insulin or transferrin receptors at the BBB resulted in a comparable brain uptake of 1–3% of the injected dose. This relates to the abundance of these receptors at the BBB, which is comparable in humans [46]. However, the abundance of the mouse BBB transferrin receptor is approximately 7-fold higher than that of the mouse BBB insulin receptor [46,47]. Therefore, targeting the mouse BBB insulin receptor would produce lower levels of brain uptake. The manufacturing of IgG fusion proteins presents advantages compared to chemical conjugation, including simplified downstream purification due to protein-A capture [48]. IgG fusion proteins were engineered targeting both transferrin and insulin BBB receptors with high affinities in the low nM range (Table 1 and Table 2). The extensive number of peer-reviewed publications discussed below validated the high-affinity approach for the transport across the BBB, targeting either the insulin or the transferrin receptor.

There are, however, few publications postulating that a low-affinity monovalent MAb directed to the BBB TfR transport system may result in improved brain uptake [49,50]. This is based on the hypothesis that bivalent TfRMAbs cause TfR clustering and selective triage of the antibody-TfR complex to the lysosome and degradation of TfR on the cell membrane, whereas this is avoided with monovalent TfRMAbs [49,50]. However, this was based on tissue culture experiments with TfRMAb-avidin fusion proteins, which are known to form tetrameric structures from the association of avidin monomers [51,52]. No toxic effects of other high-affinity TfRMAb fusion proteins were reported in in vitro or in vivo studies. Chronic treatment with intravenous (IV) 2 mg/kg BW TfRMAb-GDNF twice weekly for 12 weeks produced no downregulation of the BBB TfR, as the terminal pharmacokinetics and brain uptake were comparable to those obtained prior to the chronic treatment [53]. Moreover, no evidence of BBB TfR downregulation was reported in a chronic study performed in the cynomolgus monkey with pabinafusp alfa, the high-affinity human TfRMAb-IDS fusion protein, with doses up to 30 mg/kg/week for 26 weeks [54]. Kinetics modeling of the receptor-mediated transport across the BBB showed that the optimal receptor-binding properties would be an MAb with a KD of 0.5–5 nM and an association rate constant (kon) of 10^5^–10^6^ M^−1^ s^−1^, which would produce a dissociation T_1/2_ of ~10–120 min [55]. Targeting MAbs, i.e., TfRMAb and/or HIRMAb, with these kinetic properties produced therapeutic brain delivery at a low injection dose of 1–3 mg/kg BW in the various CNS models discussed below, including clinical trials in LSD.

The brain uptake via a BBB receptor-mediated transport is a function of the antibody affinity for the receptor, the injection dose, and the plasma area under the curve (AUC), which may be affected by the therapeutic domain of the fusion protein, as in the case of LSD enzymes targeting peripheral M6P receptors. For example, the fusion of IDUA to the transport MAb reduces the brain AUC of the fusion protein compared to the MAb alone [56]. Kinetics modeling showed that the lower the affinity of the antibody for the TfR, the greater the ID required to maintain a given brain AUC [55]. For example, the brain AUC of a TfRMAb-IDUA fusion protein with a moderate affinity for the TfR, KD = 36 nM, would require an injected dose of 30 mg/kg BW to produce a brain AUC comparable to the one of a TfRMAb-IDUA fusion protein with high affinity (KD = 0.36–3.6 nM) at a 10-fold lower injected dose of 3 mg/kg BW [55]. A lower therapeutic dose is also preferred to reduce potential adverse effects, as in the case of IgG-neurotrophic factor fusion proteins [31,57].

## 3. Enzyme-IgG Fusion Proteins

Most of the lysosomal storage disorders (LSD) affect the CNS, causing neurologic manifestations such as mental retardation and neurodegeneration [58,59]. The treatment of LSD is possible with enzyme replacement therapy (ERT). However, ERT is unable to treat the brain, as these large proteins do not cross the BBB [60,61]. The re-engineering of these enzymes as brain-penetrating IgG fusion proteins represents a potential solution for the treatment of LSD. Today, the genetic engineering of several IgG-enzyme fusion proteins has been reported (Table 1 and Table 2). These fusion proteins were designed for the treatment of a variety of LSDs, and their corresponding bifunctionality was validated biochemically and in experimental animals as well as in clinical trials. This technology was also validated for other potential therapeutics for the CNS, including decoy receptors, bispecific MAbs, and neurotrophins (Table 1 and Table 2).

### 3.1. HIRMAb-Iduronidase (HIRMAb-IDUA)

A brain-penetrating iduronidase (IDUA), an enzyme that is mutated in Hurler’s MPSI syndrome [58], was completed by the insertion of the cDNA of the mature human IDUA (GenBank NP_00194), minus the 26-amino-acid signal peptide at the C-terminus of the heavy chain of the HIRMAb via a short Ser-Ser linker (Table 1) [22]. The HIRMAb-IDUA fusion protein maintained the affinity for the targeting insulin receptor in the low nM range, and the IDUA enzyme activity was comparable to the specify activity of the of the recombinant IDUA [22]. The fusion protein targeted the lysosomal compartment in Hurler fibroblasts and markedly reduced the accumulation of glycosaminoglycans (GAG) in these cells [22]. The biodistribution of the HIRMAb-IDUA was investigated in the rhesus monkey using radio-iodinated material and compared to that of the recombinant human IDUA (Aldurazyme) [62]. The quantitative whole-body autoradiography confirmed the transport of the fusion protein across the BBB, showing a global biodistribution of the HIRMAb-IDUA throughout the brain (Figure 3). On the contrary, recombinant IDUA did not penetrate the brain through the BBB, showing background activity in the primate brain (Figure 3). The levels of brain uptake of the HIRMAb-IDUA approximated 1% of injected dose (ID) [22,62]. The biodistribution of the HIRMAb-IDUA fusion protein in the peripheral tissues was comparable to that of the recombinant IDUA, as both were taken up in the peripheral organs through the mannose-6-phosphate (M6P) receptor [62]. In addition, a significant increase in the uptake of the HIRMAb-IDUA fusion protein was observed in the vertebral bodies and joints [62]. Taking into consideration that the normal enzyme activity of IDUA in a human brain ranges from 0.5 to 1.5 units/mg of protein [63], it may be possible to normalize the brain IDUA in a Hurler individual with the administration of 1 mg/kg BW of the fusion protein, which may result in a brain concentration of 3.0 ng/mg of brain protein or 1.1 units of IDUA enzyme activity per mg of brain protein [22].

The efficacy of the IgG-IDUA fusion protein was investigated in a mouse model of MPSI using a subrogate fusion protein that comprised the mouse TfRMAb fused to the mouse IDUA (Table 2) [35]. Six-month-old MPSI mice were treated with 1 mg/kg BW TfRMAb-IDUA IV twice weekly for 8 weeks. Electron microscopy showed a marked reduction in lysosomal inclusion bodies in animals treated with brain-penetrating IDUA fusion protein compared with saline (Figure 4) [35]. The administration of the TfRMAb-IDUA produced a marked reduction in GAG in the peripheral organs that was comparable to that reported for the recombinant IDUA [35].

The HIRMAb-IDUA (valanafusp alpha) was the first brain-penetrating IgG fusion protein that completed a phase I/II clinical trial in Hurler MPSI [64,65]. Pediatric MPSI patients treated with laronidase were switched to valanafusp alpha and treated with weekly IV infusions of 1, 3, or 6 mg/kg for 6 months. Patients completing the study remained in the extension arm for another 6 months [65]. A dramatic improvement in somatic parameters was described in the valanafusp-alpha-treated patients for 52 weeks, including 23% and 26% reductions in liver and spleen volumes, respectively, compared with the baseline levels [65]. The improvement in the somatic parameters was attributed to the dual targeting of valanafusp alpha in the peripheral organs via both the insulin and M6P receptor [65]. In addition, there was a significant increase in shoulder flexion (10.9°) and extension (9.5°) following treatment with valanafusp alpha for 26 weeks [65], which may be related to the preferential targeting of the HIRMAb-IDUA in the vertebral bodies and joints that was observed in primates [62]. The treatment of severe and attenuated MPSI patients with valanafusp alpha resulted in a mean improvement across all cognitive domains [65].

### 3.2. HIRMAb-Iduronate 2-Sulfatase (HIRMAb-IDS)

Iduronate 2-sulfatase (IDS) is the lysosomal storage enzyme that is mutated in Hunter’s MPSII syndrome [66]. A brain-penetrating form of IDS was engineered as HIRMAb-IDS fusion protein using a similar strategy as the one used in the production of HIRMAb-IDUA described above in Section 3.1 (Table 1) [23]. The cDNA of the mature human IDS (GenBank NP_000193), minus the 25-amino-acid signal peptide, was fused at the C-terminus of the heavy chain of the HIRMAb via a short Ser-Ser linker. The HIRMAb-IDS fusion protein, expressed in either COS or CHO cells, maintained the affinity for the targeting insulin receptor in the low nM range, and the IDS enzyme activity was comparable to the specific activity of the recombinant IDS, Elaprase [23]. The HIRMAb-IDS fusion protein targeted the lysosomal compartment in Hunter MPSII fibroblasts, and it was able to reduce the accumulation of GAG [23,67]. The biodistributions of both the HIRMAb-IDS fusion protein and Elaprase were investigated in the rhesus monkey using Bolton–Hunter-iodinated material [68]. The film autoradiography of the primate brain confirmed a global distribution of the fusion protein, whereas the non-brain-penetrating Elaprase showed background activity (Figure 5) [68]. The brain uptake of the HIRMAb-IDS fusion protein approximated 1% ID/primate brain [68]. The organ uptake ratio of HIRMAb-IDS:Elaprase was 38-fold higher in the brain, as IDS does not cross the BBB, whereas in the peripheral tissues, this ratio was near 1, as both proteins are taken up via the M6P receptor [68]. The brain uptake estimate following the administration of 1 mg/kg BW in humans was projected to be 0.34 U/mg protein, which may produce a therapeutic effect in the brains of MPSII patients [67,68]. The safety and dose ranging study for the HIRMAb-IDS fusion protein was completed in patients with Hunter syndrome; however, the results have not yet been published (NCT02262338). A mouse surrogate molecule with the anti-mouse TfR as the transport domain and human IDS as the therapeutic domain was also produced and validated in mice, wherein the brain uptake was high and comparable to the human fusion protein at 1.3% ID/mouse brain [36].

The production of another brain-penetrating IgG-IDS fusion protein was also reported [24]. This fusion protein is similar to the HIRMAb-IDS described above but is directed to the human TfR. The TfRMAb-IDS, later designated pabinafusp alfa, was used to complete phase I/II clinical trials in Hunter MPSII patients in Japan and Brazil [69,70] and a phase II/III trial in Japan [71]. It was reported that the pabinafusp alfa, dosed at 2 mg/kg BW for 52 weeks in MPSII subjects, produced a significant reduction in the levels of heparan sulfate (HS) in the cerebrospinal fluid (CSF), which was used as the primary efficacy endpoint [71]. Evaluations of neurocognitive developments, used as the secondary end point, showed positive changes in the age-equivalent score in subjects with attenuated MPSII and in the initial phase of the severe subtype but not in severe MPSIII patients in the middle and late stages of the disease [71]. The positive effect of the fusion protein in the peripheral end points, i.e., serum HS and liver and spleen volumes, was comparable to that of the IDS enzyme replacement therapy [71]. Based on this trial, pabinafusp alfa was approved by the Ministry of Health, Labour and Welfare (MHLW) in Japan for the treatment of Hunter MPSII syndrome [72].

### 3.3. HIRMAb-Arylsulfatase A (HIRMAb-ASA)

A mutation of the Arylsulfatse A (ASA) gene causes the lysosomal storage disorder metachromatic leukodystrophy (MLD) [73]. Since ASA does not cross the BBB, it may be possible to treat the brain in MLD with a brain-penetrating IgG-ASA fusion protein. The genetic engineering of the HIRMAb-ASA fusion protein was performed using the standard recombinant technology strategy, wherein cDNA of the mature human ASA (GenBank NP_000478), minus the signal peptide, was fused at the C-terminus of the heavy chain of the HIRMAb via a short Ser-Ser linker (Table 1) [25]. The HIRMAb-ASA fusion protein maintained a high affinity for the targeting insulin receptor in the low nM range and high ASA enzyme activity [25]. Confocal microscopy showed that the ASA fusion protein is triaged to the lysosomal compartment. The biodistribution of the HIRMAb-ASA fusion protein in the rhesus monkey showed a global distribution in brain, with a brain uptake of 1.1% ID/primate brain [25]. Based on this finding, the brain levels of HIRMAb-ASA were predicted to be 14 ng/mg protein following the administration of 2.5 mg/kg in humans [25]. This represents 14% of the levels of ASA in the normal human brain [74], and it may be sufficient to treat this lysosomal storage disorder, as the replacement of just 1–2% of the normal enzyme activity is expected to be therapeutic [75]. This is also supported by the fact that 5–20% of the population has ASA pseudo-deficiency, with just 3–8% of the normal levels of ASA enzyme activity in brain and no MLD symptoms [76].

### 3.4. HIRMAb-N-Sulfoglucosamine Sulfohydrolase (HIRMAb-SGSH)

The Sanfilippo type A syndrome or MPSIIIA is caused by a mutation of the N-sulfoglucosamine sulfohydrolase (SGSH) gene [77]. It is possible to treat the brain in MPSIIIA with a brain-penetrating IgG-SGSH fusion protein, as SGSH does cross the BBB. The genetic engineering of the HIRMAb-SGSH fusion protein was completed as described above for other enzymes, wherein cDNA of the mature human SGSH (GenBank NP_000190), minus the signal peptide, was fused at the C-terminus of the heavy chain of the HIRMAb via a short Ser-Ser-Ser-Ser linker (Table 1) [26]. The HIRMAb-SGSH fusion protein maintained a high affinity for the targeting insulin receptor and SGSH enzyme activity near 100% of the one of the recombinant SGSHs [26]. The brain uptake in the rhesus monkey approximated 1% ID/primate brain [26]. The brain level of SGSH enzyme activity was predicted to be 0.25 U/g of brain following the administration of 3 mg/kg HIRMAb-SGSH [26]. This is comparable to the normal endogenous levels of SGSH in the brain [78], suggesting that it is possible to achieve therapeutic levels of SGSH in the MPSIIIA brain following the administration of the HIRMAb-SGSH fusion protein. The efficacy of the IgG-SGSH fusion protein was investigated in a mouse model of MPSIIIA using a subrogate fusion protein comprising the mouse TfRMAb fused to the human SGSH (Table 2) [37]. Two-week-old MPSIIIA mice were treated three times per week for 6 weeks with intraperitoneal (IP) 5 mg/kg of the TfRMAb-SGSH fusion protein or the isotype control. Studies in mice demonstrated that the administration 5 mg/kg BW IgG fusion protein IP is equivalent to the IV injection of 1 mg/kg BW [79]. High plasma levels of HIRMAb were also reported in the rhesus monkey following subcutaneous (SC) administration [80]. Mice were euthanized 1 week after the last dose of either the control or the test article [37]. HS was measured in the brain and liver by LC-MS following enzymatic digestion into disaccharides using HS disaccharide standards [37]. MPSIIIA animals treated with saline showed 30- and 36-fold elevations in HS in the brain and liver, respectively, compared to the wild-type animals (Figure 6). Treatment with the TfRMAb-SGSH reduced the levels of HS by 70% in the brain and by 85% in the liver, whereas the isotype control had no effect [37]. The data confirmed that the administration of brain-penetrating IgG-SGSH fusion protein is able to reduce the accumulation of HS in the MPSIIIA brain and a peripheral organ.

### 3.5. HIRMAb-α-N-Acetylglucosaminidase (HIRMAb-NAGLU)

The Sanfilippo type B syndrome or MPSIIIB is caused by a mutation of the α-N-acetylglucosaminidase (NAGLU) gene [81]. Since NAGLU does cross the BBB, it is possible to treat the brain in MPSIIIB with a brain-penetrating IgG-NAGLU fusion protein. The genetic engineering of the HIRMAb-NAGLU fusion protein was constructed as described above for other enzymes, wherein cDNA of the mature human NAGLU (GenBank NP_000263), minus the signal peptide, was fused at the C-terminus of the heavy chain of the HIRMAb via a short Ser-Ser-Ser-Ser linker (Table 1) [27]. However, this fusion protein showed poor levels of expression in COS cells [27]. This problem was solved by the introduction of a 31-amino-acid extended linker, corresponding to the hinge region of IgG3 [27]. The HIRMAb-NAGLU fusion protein with the extended linker was produced in CHO cells and showed a high affinity for the targeting insulin receptor. The NAGLU enzyme activity was comparable to that of the recombinant NAGLU [27]. The biochemical properties of the HIRMAb-NAGLU fusion protein were confirmed by SDS-PAGE, human IgG and NAGLU Western blot analysis, the uptake in MPSIIIB fibroblasts, and the reduction in GAG in MPSIIIB fibroblasts [27]. The brain uptake in the rhesus monkey was 1% ID/primate brain [27]. The brain level of NAGLU enzyme activity was predicted to be 0.36 U/mg of brain protein following the administration of 1 mg/kg HIRMAb-NAGLU, which is comparable to the NAGLU enzyme activity in the normal brain [82]. The data suggest that it is possible to achieve therapeutic levels of NAGLU in the MLDIIIB brain following the administration of the HIRMAb-NAGLU fusion protein.

### 3.6. HIRMAb-Acid Sphingomyelinase (HIRMAb-ASM)

A mutation of the acid sphingomyelinase (ASM) gene causes Niemann–Pick disease type A (NPDA) [83]. Since ASM does cross the BBB, it is possible to treat the brain in NPDA with a brain-penetrating IgG-ASM fusion protein. The genetic engineering of the HIRMAb-ASM fusion protein was designed as described above for other enzymes, with the exception that the ASM gene was fused to the light chain of HIRMAb in lieu of the heavy chain (Figure 2) (Table 1) [28]. Thef of the enzyme genes to the C-terminus of the heavy chain of an MAb places the enzyme in a dimeric configuration, as opposed to the monomeric configuration obtained by fusion to the light chain of an MAb. ASM forms a heterodimer with saposin C [84], so it was fused to the C-terminus of the light chain to provide a more flexible configuration (Figure 2) [28]. The cDNA of the mature human ASM (GenBank NP_000534), minus the signal peptide, was fused at the C-terminus of the light chain of HIRMAb via the 31-amino-acid extended linker corresponding to the hinge region of IgG3 [28]. The HIRMAb-ASM fusion protein was produced in COS or CHO cells and showed a high affinity for the targeting insulin receptor and high ASM enzyme activity [28]. Assuming a brain uptake of 1% ID for the HIRMAb-ASM, the administration of 3 mg/kg BW of the fusion protein produces a brain concentration of 1.5 mg/brain, which represents a therapeutic enzyme level in the brain of an NPDA mouse [85].

### 3.7. HIRMAb-Hexoaminidase A (HIRMAb-HEXA)

A mutation of the Hexoaminidase A (HEXA) gene produces Tay–Sachs disease (TSD) [86]. It is possible to treat the brain in TSD with a brain-penetrating IgG-HEXA fusion protein, as the recombinant HEXA does not cross the BBB. The genetic engineering of the HIRMAb-HEXA fusion protein was designed as described above for ASM [28]. The HEXA gene was fused to the light chain of HIRMAb to place this enzyme in a monomeric configuration (Figure 2) (Table 1) [28] and was able to form a heterodimer complex with the GM2 activator protein [87]. The cDNA of the mature human HEXA (GenBank NP_000511), minus the signal peptide, was fused at the C-terminus of the light chain of the HIRMAb via the 31-amino-acid extended linker [28]. The HIRMAb-HEXA fusion protein was produced in COS or CHO cells and showed a high affinity for the targeting insulin receptor and a high HEXA enzyme activity comparable to the recombinant HEXA protein [28]. Assuming a brain uptake of 1% ID for the HIRMAb-HEXA, the administration of 3 mg/kg BW of the fusion protein may produce a brain concentration of 2.5 U/brain, which represents a therapeutic enzyme level in the TSD brain [88].

### 3.8. HIRMAb-Palmitoyl-Protein Thioesterase (HIRMAb-PPT1)

Batten type 1 disease, or neuronal ceroid pilofuscinosis type 1 (CLN1) disease, is produced by a mutation of the palmitoyl-protein thioesterase (PPT1) [89]. In order to produce a brain-penetrating IgG-PPT1 fusion protein, the cDNA of the mature human PPT1 (GenBank NP_000301), minus the signal peptide, was fused at the C-terminus of the heavy chain of the HIRMAb, as described above for other fusion proteins, using the short Ser-Ser-Ser linker (Table 1) (Figure 2) [28]. This construct places the therapeutic domain in a dimeric configuration, as in the native PPT1 (Figure 2). However, the production the HIRMAb-PPT1 with the short Ser linker generated a fusion protein with suboptimal enzyme activity [28]. This problem was solved by the introduction of the flexible 31-amino-acid extended linker, which allowed the production of a fusion protein with a high PPT1 enzyme activity, maintaining its affinity for the insulin receptor in the low nM range [28]. Assuming a brain uptake of 1% ID for the HIRMAb-PPT1, the administration of 3 mg/kg BW of the fusion protein may produce a brain concentration of 2.6 U/brain, which represents 2.2% of the normal endogenous PPT1 activity; however, it would be therapeutic, as just 0.5% of the endogenous PPT1 activity is needed to reverse the neuropathology of CLN1 [90].

### 3.9. HIRMAb-β-Galactosidase (GLB1) (HIRMAb-GLB1)

The gene mutated in GM1-gangliosidosis is β-galactosidase (GLB1) [91]. In order to reformulate GLB1 into a brain-penetrating IgG fusion protein, the cDNA of the mature human GLB1 (GenBank NP_000395), minus the signal peptide, was fused at the C-terminus of the heavy chain of the HIRMAb, as described above for other fusion proteins, using the short Ser-Ser-Ser linker (Table 1) (Figure 2) [28]. This construct places the therapeutic domain in dimeric configuration, as in the native GLB1 (Figure 2). However, as described above for the HIRMAb-PPT1, the production the HIRMAb-GLB1 with the short Ser linker produced a fusion protein with a marked decrease in its specific enzyme activity [28]. The GLB1 enzyme activity of the HIRMAb-GLB1 fusion protein was restored by the introduction of the flexible 31-amino-acid extended linker (Table 1) [28]. This fusion protein also maintained its affinity for the insulin receptor in the low nM range [28]. Assuming a brain uptake of 1% ID for the HIRMAb-GLB1, the administration of 3 mg/kg BW of the fusion protein may produce a brain concentration of 256 U/g brain, which may represent a therapeutic GLB1 enzyme level in the brain [92]. In an attempt to validate the efficacy of the IgG-GLB1 fusion protein, a subrogate fusion protein comprrising the mouse TfRMAb fused to the human GLB1 was produced and tested in a mouse model of GM1-gangliosidosis [93]. The TfRMAb-GLB1 fusion protein was able to increase the GLB1 enzyme activity in the liver by 20%; however, it failed to increase the GLB1 activity in the brain or reduce the ganglioside content [93]. Since this surrogate fusion protein was engineered with a short Ser linker [93] that is known to produce suboptimal levels of GLB1 [28], a negative conclusion on brain effects is premature at the present time. Further studies with a surrogate mouse GLB1 construct with optimal GLB1 enzyme activity, i.e., engineered with an extended linker [28], may be needed to clarify this matter.

## 4. Bispecific Therapeutic Antibodies

Monoclonal antibodies (MAbs) are potential therapies for the CNS, i.e., in Alzheimer’s disease (AD), if they are re-engineered to cross the BBB. Novel strategies for the immune therapy of AD have been proposed with the design of bispecific MAbs [29,94]. Some designs propose the use of low-affinity MAbs for the transport domain of the fusion MAb due to limitations in the technology that is used, i.e., knobs-into-holes, which produces monovalent antibodies composed of two heterologous half-antibody molecules for either the transport or the therapeutic domain [50]. However, as discussed above in Section 2, a low-affinity transport domain presents no advantages over the high-affinity targeting MAb. Other designs maintain the bivalency of both domains, resulting in a tetravalent bispecific MAb fusion protein with a high affinity for both the transport and therapeutic domain (Figure 7) [29,94].

For example, an anti-Aβ MAb was re-engineered to cross the BBB in both directions for the immune therapy of AD [29]. This process involves three-steps: (i) the transport of the anti-Aβ antibody from the blood to the brain across the BBB; (ii) the binding to and disaggregation of Aβ fibrils in the brain; and (iii) the efflux of the anti-Aβ antibody from the brain back into the blood. This trifunctional molecule, designated HIRMAb-Aβ-ScFv, comprises (i) the transport domain, i.e., the HIRMAb; (ii) the therapeutic domain, i.e., a single-chain anti-Aβ antibody monomer (ScFv) fused to the carboxyl terminus of the heavy chain of the HIRMAb; and (iii) the binding site for the neonatal Fc receptor or FcRn, located at the CH2-CH3 interface of the human IgG constant region (Figure 7), which mediates the brain efflux of the HIRMAb-Aβ-ScFv [29]. The HIRMAb-Aβ-ScFv fusion bifunctional antibody was engineered by the insertion of an scFv directed to the Aβ^1–28^ peptide at the C-terminus of the heavy chain of HIRMAb via a short Ser-Ser linker (Table 1) [29]. The tetravalent bifunctional Mab maintained a high affinity for both Aβ and the BBB insulin receptor [29]. The pharmacokinetics and biodistribution of the HIRMAb-Aβ-ScFv fusion Mab were investigated in a rhesus monkey using an [^125^I]-labeled test article and compared with a control article that comprised the [^3^H]-labeled original murine MAb directed to Aβ (Mab-Aβ) (Figure 8) [29]. Following administration, there was no measurable decrease in the blood concentration of the control MAb-Aβ, as MAbs do not target any organ remaining in the blood compartment (Figure 8). On the contrary, there was a rapid clearance of the [^125^I]-HIRMAb-Aβ-ScFv fusion antibody from blood (Figure 8), as this fusion protein targets the brain and peripheral organs expressing the insulin receptor [29]. Thus, there was a global distribution of the fusion MAb in the brain with a preferential uptake in the gray matter relative to the white matter. The capillary depletion technique showed a high brain volume of distribution (VD) of the bifunctional fusion MAb, demonstrating that the HIRMAb-Aβ-ScFv was transcytosed across the BBB into the postcapillary brain compartment (Figure 8) [29]. On the other hand, the control mouse MAb-Aβ had a brain VD of 10 μL/g of brain, which approximates the arterial blood volume of the brain, confirming that the control MAb-Aβ does not cross the BBB, remaining in the primate blood compartment (Figure 8) [29].

In order to validate the bifunctional HIRMAb-Aβ-ScFv in a mouse model of AD, a surrogate molecule targeting the mouse TfR was engineered (Table 2) [38]. This molecule was similar to the one shown in Figure 7, with an identical therapeutic domain that comprised the ScFv to the Aβ^1–28^ peptide but with an anti-mouse TfR as the transport domain [38]. This fusion protein maintained a high affinity for both Aβ and the mouse TfR and produced a brain uptake of 3.5% [38]. The administration of 1 mg/kg BW TfRMAb-Aβ-ScFv IV three times per week or 5 mg/kg BW SC daily for 12 weeks to B6C3-Tg(APPswe, PSEN1dE9)85 Dbo/J (PSAPP) double transgenic mice produced a 40–61% reduction in the brain concentration of Aβ^1−42^ [95,96] without brain microhemorrhage [97], a common adverse side effect seen in the immune therapy of AD. A reverse construct wherein the transport domain is a form of an ScFv fused to the C-terminus of the light chain of a therapeutic MAb has also been reported [98]. These constructs present the advantage of re-engineering any therapeutic MAb into a brain-penetrating tetravalent bispecific MAb targeting either the BBB-TfR [98] or the BBB-HIR [99].

## 5. Decoy Receptor–IgG Fusion Proteins

Other potential new therapeutics for brain disorders are decoy receptors. A decoy receptor is formed by the extracellular domain (ECD) of the molecule of interest to be inactivated fused to the to the amino terminus of the Fc region of human IgG1 for stability and to facilitate downstream protein production and purification. A good example is the ECD of the tumor necrosis factor (TNF) receptor (TNFR) type II:Fc fusion protein, etanercept [100]. The TNFR decoy receptor is used to suppress inflammatory reactions in non-brain tissues [101]. TNFα has also been involved in disorders of the CNS, including stroke [102], traumatic brain and spinal cord injury [103,104], neurodegeneration [105], and depression [106]. Therefore, the production of a brain-penetrating TNFR decoy receptor may provide a treatment for these pathologies of the CNS. A model of such a protein was engineered by the insertion of the cDNA encoding the human TNFR ECD to the C-terminus of the heavy chain of the HIRMAb via a Ser-Ser linker (Table 1) [30], as described above in Section 4 for the tetravalent bispecific MAb. This construct produced in CHO cells maintained a high affinity for the BBB insulin receptor and TNFα [30]. The brain uptake of the HIRMAb-TNFR was investigated in the rhesus monkey and compared with that of the TNFR:Fc. The HIRMAb-TNFR fusion protein was transported across the BBB, producing a brain uptake of 3% ID [30]. On the other hand, the non-brain-penetrating TNFR:Fc produced a brain uptake comparable to that of IgG1, which is confined to the blood compartment in the brain (Figure 9). The ratio for the organ permeability–surface area (PS) of the HIRMAb-TNFR relative to the organ PS for the TNFR:Fc in the rhesus monkey is shown in Figure 9 (bottom). This ratio demonstrates that both HIRMAb-TNFR and TNFR:Fc are transported into peripheral organs, as the PS ratio approximates 1 (Figure 9). The PS ratio was >30 in the brain, as just the HIRMAb-TNFR is transported into the primate brain (Figure 9) [30].

A surrogate molecule was engineered to validate this construct in experimental mouse models of stroke, Parkinson’s disease (PD), and AD. This construct was produced by the insertion of the human TNFR into the C-terminus of the heavy chain of an MAb directed to the mouse BBB TfR (Table 2) [39]. The bifunctional construct maintained a high affinity for TNFα, which was comparable to that of etanercept, as well as a high binding affinity for the mouse TfR [39]. In a 6-hydroxydopamine model of PD, the mouse TfRMAb-TNFR was neuroprotective, reducing both the apomorphine- and amphetamine-induced rotation and increasing the vibrissae-elicited forelimb placing and the striatal tyrosine hydroxylase (TH) enzyme activity [39]. On the contrary, etanercept had no effect on striatal TH enzyme activity or neurobehavior, as it is not transported through the BBB [39]. In a reversible middle cerebral artery occlusion mouse stroke model, the TfRMAb-TNFR also produced neuroprotection, causing a significant reduction in the hemispheric, cortical, and subcortical stroke volumes and neuronal deficit, whereas etanercept had no effect [107]. In a mouse model of AD, chronic treatment with TfRMAb-TNFR, but not with either saline or etanercept, produced a marked reduction in neuroinflammation and in both Aβ peptide and plaque load and improved recognition memory [108]. As observed with the TfRMAb-Aβ-ScFv [38], no sign of microhemorrhage was seen with the chronic treatment of TfRMAb-TNFR [108].

## 6. Neurotrophic Factor-IgG Fusion Proteins

Neurotrophic factors could potentially be developed as new treatments of brain disorders, as in the case of stroke, traumatic brain injury, or chronic neurodegeneration, such as Parkinson’s disease (PD) [109,110,111,112,113,114,115,116,117,118]. However, as discussed above for other protein-based therapeutics for the CNS, the drug development of neurotrophic factors is limited by the lack of transport of across the BBB. The engineering of brain-penetrating neurotrophic factors has been reported for erythropoietin (EPO), glial-derived neurotrophic factor (GDNF), and brain-derived neurotrophic factor (BDNF), and details are discussed below.

### 6.1. HIRMAb-Erythropoietin (EPO) (HIRMAb-EPO)

The engineering of the brain-penetrating EPO was completed by the insertion of the cDNA of the mature human EPO (GenBank NP_000790) at the C-terminus of the heavy chain of the HIRMAb using the short Ser-Ser-Ser linker (Table 1) in a tandem vector carrying both the light and heavy chain of HIRMAb [31]. The design of this fusion protein placed the EPO in dimeric configuration, as shown in Figure 2 for HIRMAb-PPT1 or HIRMAb-GLB1. The HIRMAb-EPO fusion protein expressed in COS cells demonstrated a high affinity for both the BBB insulin receptor and the EPO receptor (EPOR). The biodistribution of both EPO and HIRMAb-EPO was investigated in the rhesus monkey. The brain uptake of HIRMAb-EPO was high at 2% ID/monkey brain [31]. On the contrary, EPO, which does not cross the BBB, had a brain uptake comparable to human IgG1, which is confined to the blood compartment of the primate [31]. The mouse TfRMAb-EPO surrogate fusion protein (Table 2) was constructed to investigate the efficacy of the brain-penetrating EPO in an experimental model of stroke and PD [40]. The TfRMAb-EPO traversed the mouse BBB and had an uptake of 2% ID/mouse brain, which is similar to that of the HIRMAb-EPO in the rhesus monkey, and maintained a high affinity for the mouse BBB TfR and EPOR [40]. The mouse TfRMAb-EPO fusion protein was neuroprotective in a reversible middle cerebral artery occlusion (MACO) stroke model, dosed at 1 mg/kg BW IV following MACO. There was a significant reduction in the hemispheric stroke volume as well as in the neuronal deficit, whereas EPO had no effect [119,120]. The mouse TfRMAb-EPO fusion protein was also neuroprotective in a 6-hydroxydopamine model of PD [121]. The IV administration of 1 mg/kg BW of the fusion protein given 1 h after the toxin and every other day for 3 weeks was neuroprotective, reducing both the apomorphine- and amphetamine-induced rotation and increasing the vibrissae-elicited forelimb placing and the striatal TH enzyme activity [121]. In a model of experimental AD, this fusion protein presented therapeutic benefits on Aβ load, synaptic loss, and microglial activation as well as improved special memory and did not show evidence of microhemorrhage [122].

### 6.2. HIRMAb-Glial-Cell-Derived Neurotrophic Factor (GDNF) (HIRMAb-GDNF)

The production of a brain-penetrating IgG-GDNF fusion protein was also reported targeting either the human IR or the mouse TfR, respectively (Table 1 and Table 2) [32,41]. The mature human GDNF cDNA corresponding to amino acids Ser^78^-Ile^211^ (GenBank P39905) was fused to the C-terminus of the heavy chain of the HIRMAb of the TfRMAb using the short Ser-Ser linker [32,41]. As mentioned above, this construct placed the GDNF in a dimeric configuration, as shown in Figure 2 for HIRMAb-PPT1 or HIRMAb-GLB1. These fusion proteins, expressed in either COS or CHO cells, demonstrated a high affinity for both the corresponding target receptor and the GDNF receptor (GFRa1). The administration of [^125^I]-HIRMAb-GDNF in a rhesus monkey showed a global distribution of the fusion protein throughout the brain, with a brain clearance (CL) of 0.8 μL/min/g [123]. Conversely, [^125^I]-labeled GDNF, which is not transported across the BBB, produced a brain CL comparable to human IgG1, as both molecules remain in the blood compartment [123]. The brain uptake of the mouse fusion protein was high at 3% ID/mouse brain [41]. The mouse IgG-GDNF surrogate was neuroprotective in a 6-hydroxydopmanie model of PD [124]. The IV administration of 1 mg/kg BW of the fusion protein given 1 h after the toxin and every other day for 3 weeks was neuroprotective, reducing both the apomorphine- and amphetamine-induced rotation and increasing the vibrissae-elicited forelimb placing and the striatal TH enzyme activity [124]. The mouse TfRMAb-GDNF fusion protein was also neuroprotective in the MACO stroke model. The administration of 1 mg/kg BW IV fusion protein following MACO produced a 30% reduction in cortical stroke volume, whereas GDNF alone had no effect on stroke volume (Figure 10) [125]. Furthermore, cotreatment with TfRMAb-GDNF and TfRMAb-TNFR following MCAO enhanced neuroprotection, reducing the cortical stroke volume to 69% [125]. A study of an MAb-GDNF fusion protein targeting the human insulin receptor failed to produce neuroprotection in an MPTP (1-methyl-4-phenyl-1,2,3,6-tetrahydropyridine) model of PD in primates [126]. Animals were treated a week after the neurotoxin with 1 or 5 mg/kg BW IV twice weekly for 22 doses, and no improvements in parkinsonian motor symptoms were reported in either dose [126]. Since a BBB-penetrating GDNF was neuroprotective in the 6-hydroxydopamine mouse model of PD at the low dose of 1 mg/kg BW, it is unclear why negative results were observed in the MTPT primate model. Besides the obvious difference in animal models, additional time course and dose-finding studies in the MPTP model may be needed to clarify this matter.

### 6.3. HIRMAb-Brain-Derived Neurotrophic Factor (BDNF) (HIRMAb-BDNF)

Another neurotrophic factor that was re-engineered to cross the BBB was BDNF (Table 1) [33]. The genetic engineering was performed as described above for EPO and GDNF, by insertion of the mature cDNA of BDNF coding for amino acids His^1^-Arg^117^ into C-terminus of the HIRMAb heavy chain via a short Ser-Ser-Met linker [33]. The expression of the HIRMAb-BDNF fusion protein was performed in either COS or CHO cells using a TV coding for both the light and heavy chain of the HIRMAb-BDNF [33]. The fusion protein maintained a high affinity for both the human BBB insulin receptor and the human trkB receptor for BDNF [33]. Studies in the rhesus monkey showed that the brain VD for the [^3^H]-labeled HIRMAb-BDNF fusion protein was constant in the post-vascular supernatant measured with the capillary depletion method, demonstrating that the HIRMAb-BDNF fusion protein is transcytosed through the BBB and into brain parenchyma. Based on the specific activity of the labeled fusion protein, the brain VD was 24 ± 1 ng/g of HIRMAb-BDNF fusion protein at 3 h after injection [33]. This value is >10-fold higher than the endogenous concentration of BDNF in the adult primate brain [127], suggesting that it may be possible to reach therapeutic levels of this neurotrophic factor in brain following the administration of the BBB-penetrating HIRMAb-BDNF fusion protein.

## 7. Avidin–IgG Fusion Protein

The genetic engineering, expression, and validation of an HIRMAb-avidin (AV) fusion protein and its mouse TfRMAb-AV surrogate molecule were reported (Table 1 and Table 2) [34,42]. These avidin fusion proteins were aimed to develop a universal brain delivery system that can be adapted to a variety of mono-biotinylated drugs, including siRNA [15,34,42]. A potential concern with avidin fusion proteins is the possible immunogenicity of the chicken avidin in humans and the induction of a human anti-avidin response. However, avidin has been administered to humans in 5–10 mg doses intravenously without immunologic reactions [128,129]. These fusion proteins were generated by the insertion of the AV cDNA corresponding to amino acids Ala^1^-Glu^128^ of the mature chicken avidin protein (GenBank X05343) at the C-terminus of the heavy chain of both HIRMAb and mouse TfRMAb with a short Ser-Ser-Ser linker [34,42]. This configuration places the avidin moiety in a parallel dimer conformation (as in the case of the neurotrophic factors discussed above in Section 6), which replicates the parallel association of two avidin monomers to form a dimer [130]. The binding activity of the AV-fusion proteins for the corresponding target BBB receptor was comparable to the appropriate Mab control. The mouse AV-fusion protein showed a high brain uptake of 2% ID/mouse brain. The therapeutic efficacy of the fusion protein was demonstrated in human U87 cancer cells with a knockdown of luciferase gene expression by mono-biotinylated siRNA [15]. The potential application of brain-penetrating AV-fusion proteins as peptide radiopharmaceuticals for AD was also reported [34,131].

## 8. Safety

As valanafusp alpha (HIRMAb-IDUA) and HIRMAb-IDS entered phase I/II clinical trials in Hurler MPSI and Hunter MPSII, respectively, these fusion proteins were subjected to extensive safety evaluations. Tissue cross-reactivity studies were performed under Good Laboratory Practice (GLP) and showed comparable binding of HIRMAb fusion proteins to human and rhesus monkey organs [56,132], validating further the toxicological studies in these animals. A 6-month GLP toxicological study was conducted with HIRMAb-IDUA in 40 juvenile primates that were dosed weekly with IV infusions of up to 30 mg/kg BW for 6 months [56]. The sole adverse event was hypoglycemia at a high dose of 30 mg/kg BW [133]. This was due to a secondary pharmacologic effect related to an allosteric agonistic effect of insulin and was fully preventable by performing the infusion of the drug in dextrose-saline [133]. No evidence of chronic toxicity was observed in any primate during the 6-month treatment study, including animals euthanized after a 1-month recovery period. No significant changes were reported in physical exam, food intake, EKG, ophthalmoscopic exam, body weights, or organ weights in any of the treatment groups relative to controls [56]. The pharmacokinetics was predictable over the entire dose range. As expected, anti-drug antibodies (ADA) were generated in response to the human fusion protein in primates; however, those were not neutralizing as the end-of-study pharmacokinetics shows no change in either clearance from plasma or in plasma enzyme activity [56]. Similar GLP chronic toxicological studies were performed with the HIRMAb-IDS in rhesus monkey, with exception that the infusion of the fusion protein was conducted in dextrose-saline to prevent any potential hypoglycemic event [132]. No adverse effect or chronical toxicity were reported; thus, the no-adverse-event level (NOAEL) for the HIRMAb-IDS was set at 30 mg/kg BW [132]. The generation of ADA in HIRMAb-IDS-treated rhesus monkeys was similar to that described in the HIRMAb-IDUA toxicological study [56], with the majority of ADAs against this fusion protein directed to the HIRMAb alone [132]. Valanafusp alpha was produced under Good Manufacturing Practice and passed the safety and potency testing set up by regulatory agencies, i.e., the U.S. Food and Drug Administration and the Brazilian Health Regulatory Agency (Anvisa) [65]. In the phase I/II clinical trial with valanafusp alpha in pediatric MPS I patients, the test article was administered by infusion in 5% dextrose-saline at 1, 3, or 6 mg/kg for 6 months, followed by an extension of another 6 months. The IDUA fusion protein was well-tolerated in more than 570 infusions. There was a hypoglycemic drug-related adverse effect with an incidence of 5.9%, which was transient and resolved within 10–20 min following a snack or glucose sachet. It must be noticed that 62% of all episodes were at the high dose of 6 mg/kg, so the hypoglycemic incidence at the therapeutic dose of 1–3 mg/kg BW was just 2.8% [65]. The mean glucose was reported to be normal at 101 ± 20 mg/dL over the course of the 52 trial weeks and >3000 glycemia measurements [65]. There were 10 infusion-related reactions (IRRs) reported in this clinical trial, which represent an incidence of just 1.7%, and 60% of the IRRs were observed in a single patient that was not previously on enzyme replacement therapy and in whom tolerance to the drug developed by the 10th week [65]. There was a poor correlation between IRRs and the ADA titer [65]. The relatively low rate in IRR may be due to the presence of Tregitopes in the constant region of human IgG, which may induce immunotolerance [134]. The pharmacology and safety were also reported for the acute administration HIRMAb-GDNF in the rhesus monkey [135]. The GLP toxicological study aimed for an acute treatment of stroke was completed with IV doses of up to 50 mg/kg BW of fusion protein over a 60 h period to 56 primates [135]. No adverse events were reported in the 2-week terminal toxicology study, and no neuropathologic changes were observed either [135]. Thus, a no-observable-adverse-effect level was established in the rhesus monkey for the acute administration of the HIRMAb-GDNF fusion protein [135]. A publication claimed that the treatment with HIRMAb-GDNF caused proliferative lesions in the pancreas of four of seven animals at the low dose of 1 mg/kg but not at the high dose of 5 mg/kg in an MPTP-PD model in the rhesus monkey [126]. This observation is difficult to interpret, as there was no dose-dependent effect reported, as the high dose of the fusion protein had no adverse effect. The lesions observed in the primate pancreas are detected in 30% of all human autopsies, and they are not pre-malignant [136]. Furthermore, no pancreatic lesions were reported following 6 months of treatment with either HIRMAb-IDUA or -IDS fusion protein at doses as high as 30 mg/kg BW/week [56,132]. Another study reported no toxic effect of the surrogate mouse TfRMAb-GDNF fusion protein in IV-dosed mice at 2 mg/kg BW twice weekly for 12 weeks [53]. The chronic treatment with the mouse fusion protein caused no histologic changes in the brain and cerebellum, kidney, liver, spleen, heart, or pancreas; no change in body weight; and no change in 23 serum chemistry measurements [53]. A low-titer immune response against the fusion protein was reported, which was directed against the variable region of the antibody part of the fusion protein, with no immune response directed against either the constant region of the antibody or against GDNF. As shown for HIRMAb fusion proteins, these antibodies were not neutralizing, as no changes were reported in the pharmacokinetics and brain uptake at the end of the 12 weeks of treatment [53].

The safety for the TfR pathway has raised some concerns. A decrease in circulating reticulocytes was reported after an acute dosing of a low-affinity TfRMAb [137]. A mutation of the Fc effector function seemed to rescue the reduction in reticulocytes [137]. However, several chronic studies using high-affinity TfRMAb did not report changes in circulating reticulocytes [35,37,39,53,95,96,107,108,119,120,121,122]. Moreover, the elimination of the effector function in the TfRMAb-EPO fusion protein by the substitution of the Asn residue at position 292 of the mouse IgG1 constant region in the TfRMAb with Gly produced a mutant fusion protein with a marked increase in clearance, resulting in a several-fold reduction in Cmax following IV or SC administration compared to the wild-type molecule [57]. The data suggest that the acute effect of TfRMAbs on reticulocytes is transient and reversed by chronic treatments, and that the potential benefit of the elimination of the effector function may be offset by its rapid pharmacokinetic clearance. In addition, a chronic study with the mouse TfRMAb-EPO fusion protein in the PSAPP mouse model of AD demonstrated improved hematology safety as well as better behavioral and therapeutic indices compared with recombinant EPO alone [138]. Another study in rhesus monkeys reported toxicity associated with the chronic administration of a humanized TfRMAb [139]. Treatment with 30 mg/kg BW of this MAb caused anemia associated with suppressed blood reticulocytes. The immunohistochemistry of terminal brain tissue showed microglia activation in conjunction with astrogliosis. A moderate axonal/myelin degeneration was also reported in the sciatic nerve, suggesting that this vector may have a narrow therapeutic index [139]. Nevertheless, the authors speculated that further studies may be needed to determine if this neuropathology is induced by the antibody effector function or if it is an intrinsic property of targeting the TfR in the brain [139]. On the contrary, another chronic study performed in the cynomolgus monkey with pabinafusp alfa (human TfRMAb-IDS fusion protein) reported no effector function and no significant toxicological changes at doses up to 30 mg/kg/week for 26 weeks [54]. It is possible that discrepancies in toxic effects targeting the BBB TfR are related to the intrinsic nature of the different TfRMAbs, most likely involving the target epitope. TfRMAbs were reported to have different properties, including the inhibition of cell growth [140,141]. In a phase II-III clinical trial with pabinafusp alfa in Hunter MPSII patients, 14 of 28 subjects presented IRR that were transient and clinically manageable without the cessation of the administration of the test article. Another 14 patients developed anti-pabinafusp alfa antibodies but had no IRR. Serious adverse events in five patients were reported to be unrelated to the test drug. These included one death due to respiratory failure and resultant hypoxic encephalopathy, conditions that are associated with MPSII [71].

The production of IgG fusion proteins for the GLP toxicological studies was conducted in 50 L Wave bioreactors in perfusion mode in serum-free culture medium and with stably transfected CHO cell lines [48,56,132,133,135,139,142]. Two bioreactor volumes of the conditioned medium were collected per day over a 3–4-week period. The downstream purification process involved protein A affinity chromatography, acid hold for viral inactivation, cation and anion exchange chromatography, nanofiltration, and diafiltration [48,56,132,133,135]. The safety of the production process was validated by GLP viral clearance validation studies and by parameters set up by the FDA, including CHO host protein and DNA, residual protein A, endotoxin, and sterility [48,56,132,133,135]. IgG fusion proteins were reported to be stable in a liquid formulation at 2–8 °C for more than 2 years [48,56,132,133,135]. The process was successfully transferred to a contract manufacturing organization and scaled up 10-fold for the GMP production used in clinical trials [143].

## 9. Overview and Future Perspectives

Based on the data discussed in this review, it is possible to reformulate virtually any protein-based therapeutic into a brain-penetrating IgG fusion protein therapeutic. These IgG fusion proteins comprise a transport domain that targets BBB endogenous transporters that induce receptor-mediated transport into the brain and a therapeutic domain, which exerts its pharmacological effect in the brain following transport through the BBB (Figure 1). This technology has been reduced to practice in a broad range of potential protein-based brain treatments with MAbs directed to both the BBB insulin and transferrin receptors, respectively (Table 1 and Table 2). In the majority of these constructs, the therapeutic domain is fused to the C-terminus of the heavy chain of the transport MAb domain (Figure 2), which places the therapeutic domain of the fusion protein in a dimeric configuration and mimics the mature native structure of enzymes, neurotrophic factors, decoy receptors, and bispecific MAbs. In other examples, a more flexible configuration is preferred, wherein the therapeutic domain is inserted at the C-terminus of the light chain, as in the case of HIRMAb-HEXA and HIRMAb-ASM (Figure 2), which both form heterodimer complexes with other proteins, i.e., the GM2 activator protein and saposin C, respectively (Section 3.6 and Section 3.7). Bispecific MAbs have also been engineered by the fusion of the therapeutic domain in the form of ScFv at the C-terminus of the heavy chain of the transport MAb (Figure 7). In addition, the reverse configuration of bispecific MAbs was also reported, wherein the therapeutic MAb is in a full antibody form and the transport MAb is an ScFv configuration fused to the C-terminus of the light chain of the therapeutic MAb (Section 4). The reverse configuration presents the advantage of converting any MAb into a brain-penetrating tetravalent bispecific MAb. The IgG fusion proteins are, in general, engineered with short linkers, i.e., 2–4 Ser residues, separating the transport and therapeutic domains of the fusion protein. In some examples, the short linker produced suboptimal levels of quality attributes of the fusion protein, i.e., enzyme activity, expression, and/or stability. In such a case, these attributes were restored by the introduction of a flexible 31-amino-acid extended linker corresponding to the hinge region of IgG3, as in the case of HIRMAb-NAGLU, -HEXA, -ASM, -PPT1, and -GLB1 (Section 3).

The pharmacokinetics and biodistribution of both human and mouse surrogate IgG fusion proteins were reported in rhesus primates and mice, respectively. The pharmacokinetics of the IgG fusion proteins resemble those of small molecules with rapid clearance, as these proteins target the BBB as well as peripheral organs expressing the target receptor. The latter represents an advantage for the treatment of CNS disorders also affecting peripheral organs, such as LSD. The rapid turnover rate is also advantageous, as it reduces potential adverse side effects, as in the case of EPO. The brain uptake of human and mouse IgG fusion proteins ranged from 1 to 3 % ID/brain (Section 3, Section 4, Section 5, Section 6 and Section 7). These levels of brain uptake are comparable to those of small molecules that cross the BBB, i.e., morphine and diazepam, which produce known pharmacological effects in the brain [144,145]. Based on the brain uptake data, the brain levels of IgG fusion proteins were calculated following therapeutic doses of 1–3 mg/kg BW. These estimates were shown to produce brain levels of lysosomal enzymes that were able to normalize their CNS levels in LSD, as in the case of Hurler MPSI, Hunter MPSII, MLD, Sanfilippo MPSIIIA and MPSIIIB, Niemann–Pick A, Tay–Sachs, Batten Type 1, and GM1-gangliosidosis (Section 3). Furthermore, IgG-LSD enzyme fusion proteins were validated in experimental models of Hurler MPSI, Hunter MPSII, and Sanfilippo MPSIIIA (Section 3). In addition, a model of a BBB-penetrating tetravalent bispecific MAb directed to Ab was validated in experimental AD in mice (Section 4). IgG fusion proteins with TNFR and EPO were also effective in a mouse model of AD (Section 5 and Section 6). Finally, brain-penetrating TNFR and neurotrophic factors were neuroprotective in mouse models of PD and stroke (Section 5 and Section 6).

BBB-penetrating IgG fusion proteins have shown excellent safety profiles in mice, non-human primates, and humans, in general (Section 8). Anti-drug antibodies were reported; however, those were not neutralizing, as no pharmacokinetic changes were seen at the end of chronic treatments compared with the basal parameters. Minor infusion-related immune reactions were also reported in humans, and those were similar to those seen in standard ERT. Transient hypoglycemia was reported following the administration of IgG-LSD enzymes targeting the BBB insulin receptor, an effect that was prevented by infusing the fusion protein in dextrose-saline.

In conclusion, a broad range of brain-penetrating IgG fusion proteins have been engineered and validated in various animal models of CNS disorders. The development of IgG fusion proteins is well-advanced for the treatment of Hurler MPSI and Hunter MPSII, which culminated in positive phase I-III clinical trials and the approval of the latter by the regulatory agency in Japan. Pending further drug development, other members of the brain-penetrating IgG fusion protein family discussed here are positioned to become a new generation of pharmaceutical drugs for the treatment of human CNS disorders.

## Figures and Tables

**Figure 1 pharmaceutics-14-01476-f001:**
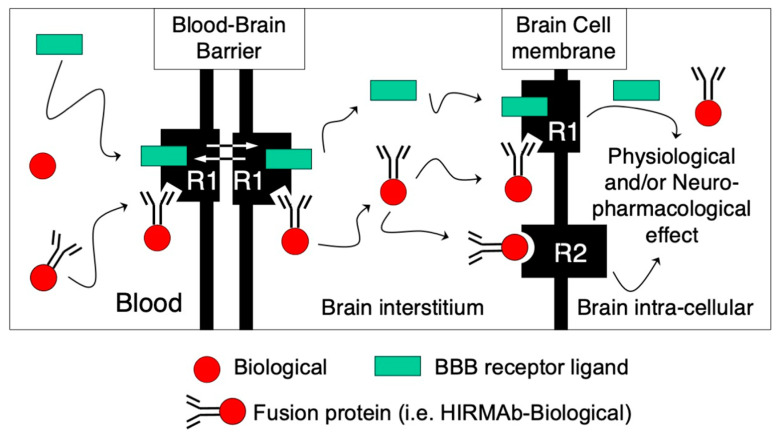
Receptor-mediated transport of IgG fusion proteins across the BBB. Biologicals (red circle) do not cross the BBB and stay in circulation following IV administration, as in the case of enzymes, MAbs, decoy receptors, and/or neurotrophic factors. These potential therapeutic agents for the CNS can be re-engineered as fusion proteins with an MAb targeting a BBB receptor that induces receptor-mediated transcytosis (R1), such as the human BBB insulin receptor (HIR) or the transferrin receptor (TfR). The transport domain of the IgG fusion protein targets the BBB R1 endogenous receptor to gain access to the brain. The transport MAb binds to an exofacial epitope of the receptor without interfering with the normal transport of its endogenous ligand (green rectangle) to gain access to the brain. Depending on the therapeutic domain of the IgG fusion protein, it can (i) bind to its ligand in the brain interstitial compartment, as in the case of bispecific MAbs or decoy receptors; (ii) target a brain cell membrane receptor (R2), such as neurotrophic factors; or (iii) be endocytosed via the same targeted R1 receptor in brain cells as lysosomal enzymes to produce physiological and/or neuropharmacological effect.

**Figure 2 pharmaceutics-14-01476-f002:**
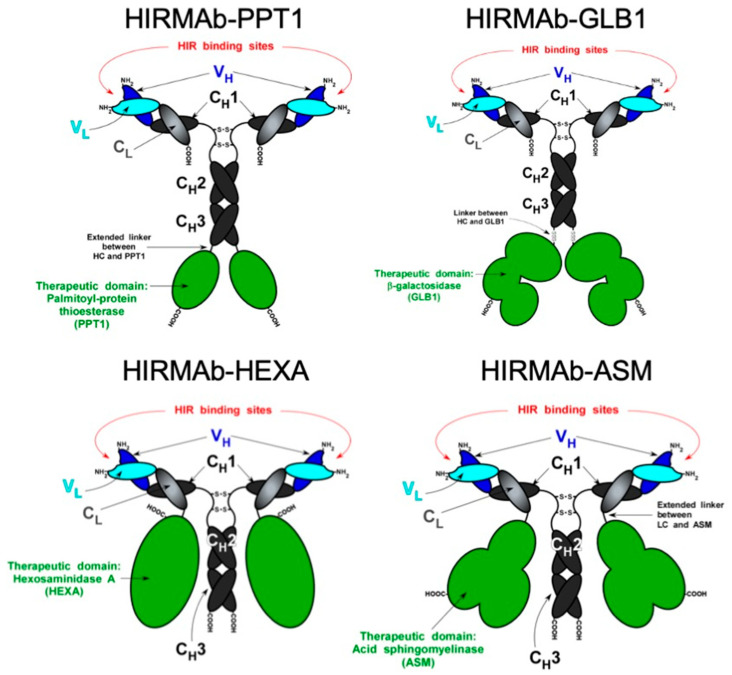
Genetic engineering of IgG fusion proteins. The therapeutic domain of the IgG bifunctional fusion protein can be fused to the C-terminus of either the heavy or light chain of the transport monoclonal antibody (MAb), in this case targeting the BBB human insulin receptor (HIR). The indication for these IgG fusion proteins is: HIRMAb-PPT1, Batten disease type 1; HIRMAb-GLB1, GM1-gangliosidosis; HIRMAb-HEXA, Tay–Sachs disease; and HIRMAb-ASM, Niemann–Pick disease types A and B. From reference [28].

**Figure 3 pharmaceutics-14-01476-f003:**
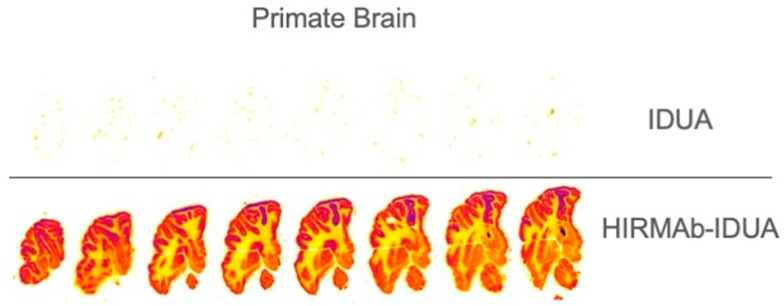
Autoradiography through eight parallel sagittal sections of the cerebral hemisphere of the rhesus monkey obtained 2 h after the IV administration of either the [^125^I]-HIRMAb-IDUA fusion protein (**bottom**) or [^125^I]-IDUA (**top**). The section on the left-hand side is the most lateral part of brain, and the section on the right-hand side is the most medial part of brain. The cerebellum is visible in the more medial sections of the brain. The BBB-penetrating HIRMAb-IDUA gained access to the brain, producing a global distribution throughout this organ. On the contrary, IDUA does not cross the BBB, showing just background activity in the primate brain. From [62] with permission.

**Figure 4 pharmaceutics-14-01476-f004:**
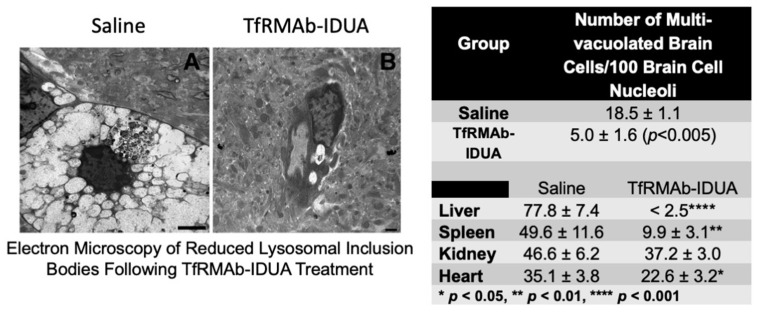
Reversal of lysosomal storage in brain of adult MPSI mice with IV injections of mouse TfRMAb-IDUA fusion protein. Six-month-old MPSI mice were treated with 1 mg/kg BW TfRMAb-IDUA IV twice weekly for 8 weeks. Electron microscopy showed a marked reduction in lysosomal inclusion bodies in animals treated with brain-penetrating IDUA fusion protein (**B**) compared with saline (**A**), resulting in a 73% reduction in brain lysosomal inclusion bodies (**right top**). The administration of the TfRMAb-IDUA produced a marked reduction in glycosaminoglycans (GAG) in the peripheral organs (**right bottom**) that was comparable to that reported for the recombinant IDUA. The data (means ± SE) are terminal organ assays at the end of the 8-week study of MPSI mice treated with either saline or 1 mg/kg BW of the TfRMAb-IDUA fusion protein. From [35] with permission.

**Figure 5 pharmaceutics-14-01476-f005:**
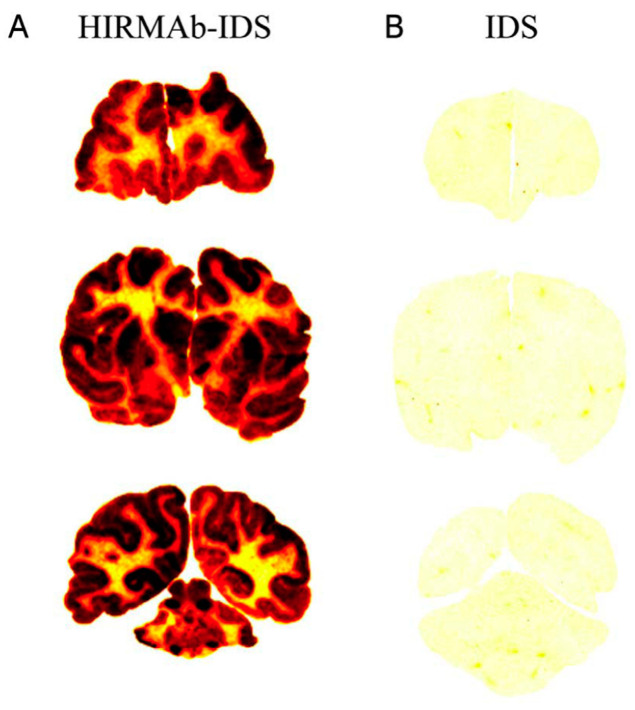
Film autoradiogram (20 µm sections) of rhesus monkey brain removed 2 h after IV injection of the HIRMAb-IDS fusion protein (**A**) or IDS (**B**). Scans were produced after labeling of the HIRMAb-IDS fusion protein or IDS with [^125^I]-Bolton–Hunter reagent. The forebrain section is on the top, the midbrain section is in the middle, and the hindbrain section with cerebellum is on the bottom. From reference [68] with permission.

**Figure 6 pharmaceutics-14-01476-f006:**
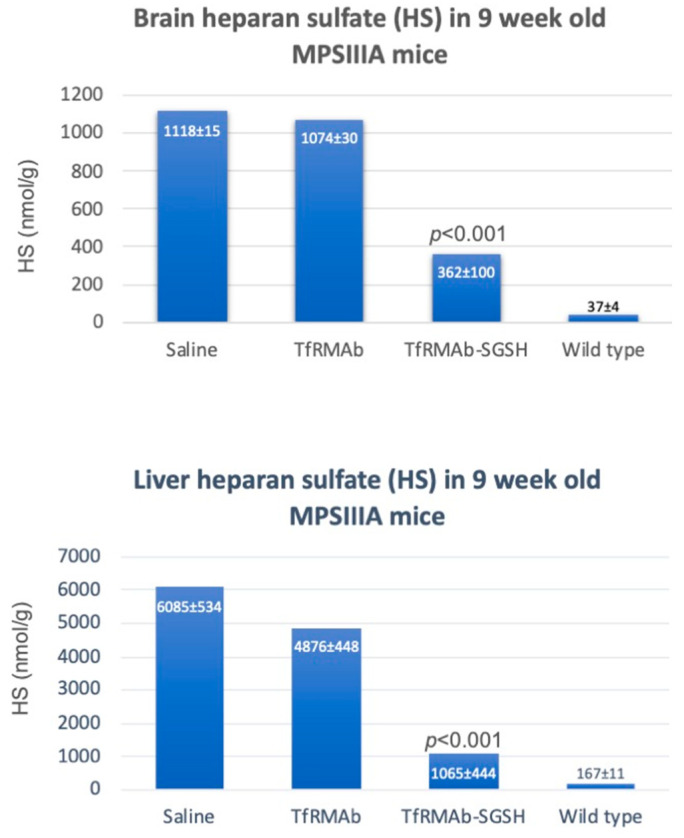
Reduction in brain heparan sulfate (HS) in the MPSIIIA mouse with systemic administration of a mouse TfRMAb-SGSH fusion protein. Two-week-old MPSIIIA mice (JAX) were treated three times per week for 6 weeks with IP 5 mg/kg of the TfRMAb-SGSH fusion protein or the isotype control (TfRMAb). The mice were euthanized 1 week after the last dose. HS was measured in brain and liver by LC-MS following enzymatic digestion into disaccharides using HS disaccharide standards. The 30-fold elevation in HS in the brain was reduced 70% by the chronic treatment with the IgG fusion protein (**top**). HS was also elevated in liver, and treatment with the mouse TfRMAb-SGSH reduced hepatic HS by 85% (**bottom**). Data are expressed as means ± SD (*n* = 8 mice/group). From [37] with permission.

**Figure 7 pharmaceutics-14-01476-f007:**
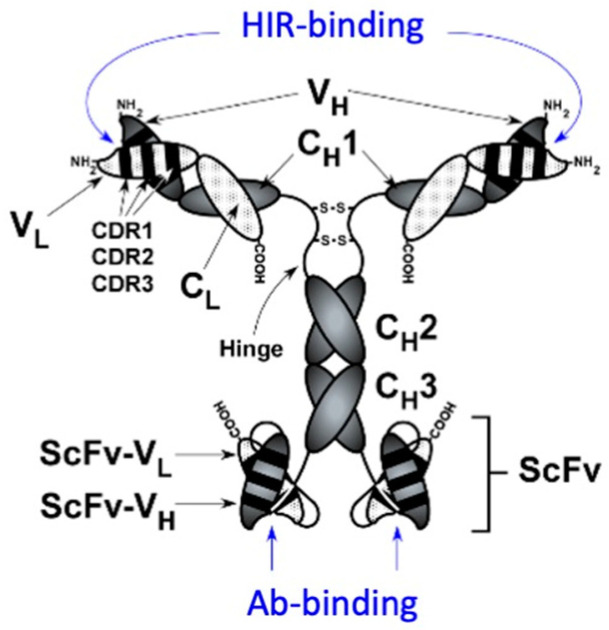
Schematic representation of a tetravalent bispecific MAb. In this construct, the transport domain of the fusion protein targets the BBB human insulin receptor (HIR), and the therapeutic domain is a single-chain anti-Aβ antibody monomer (ScFv) fused to the carboxyl terminus of the heavy chain of the HIRMAb. This tetravalent bispecific Mab maintains a high affinity for both Aβ and the BBB insulin receptor [29].

**Figure 8 pharmaceutics-14-01476-f008:**
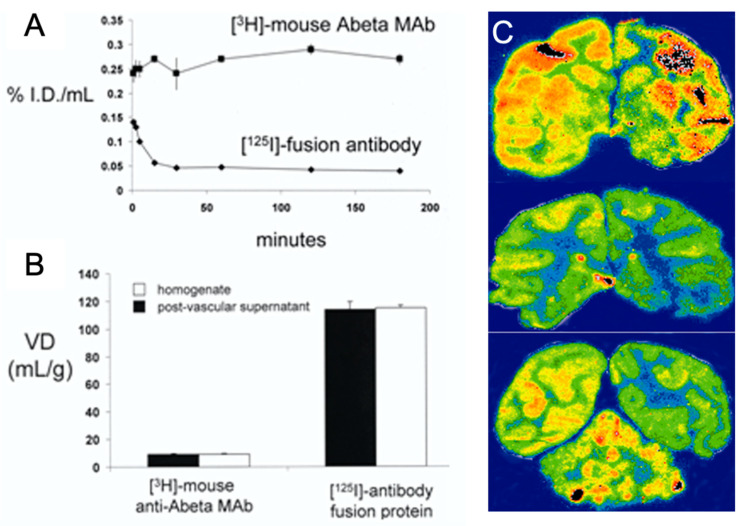
Pharmacokinetics and brain uptake of a tetravalent bispecific MAb fusion protein in the rhesus monkey. The structure of the tetravalent bispecific MAb fusion protein is shown in Figure 7. This fusion protein, designated [^125^I]-fusion antibody in this figure, comprises the transport domain, which targets the BBB human insulin receptor, and the therapeutic domain, which is a single-chain anti-Aβ antibody monomer (ScFv). (**A**) Plasma pharmacokinetics analysis showing no measurable clearance from the blood of the control [^3^H]-mouse Abeta MAb, whereas the [^125^I]-fusion antibody is rapidly cleared from blood due to uptake via the insulin receptor. (**B**) Brain VD for the [^125^I]-fusion antibody is >100 μL/g in both the brain homogenate and the post-vascular supernatant, which indicates the [^125^I]-fusion antibody is transported across the BBB. The VD for the [^3^H]-mouse Abeta MAb, 10 μL/g, is equal to the brain arterial blood volume, which indicates this antibody is not transported across the primate BBB in vivo. (**C**) Global distribution of fusion antibody to primate brain. Brain scans of adult rhesus monkey at 3 h after the intravenous administration of the [^125^I]-fusion antibody demonstrates the widespread distribution of the fusion antibody into the primate brain in vivo. The top scan is the most frontal part of brain, and the bottom scan is the most caudal part of brain and includes the cerebellum. From [29] with permission.

**Figure 9 pharmaceutics-14-01476-f009:**
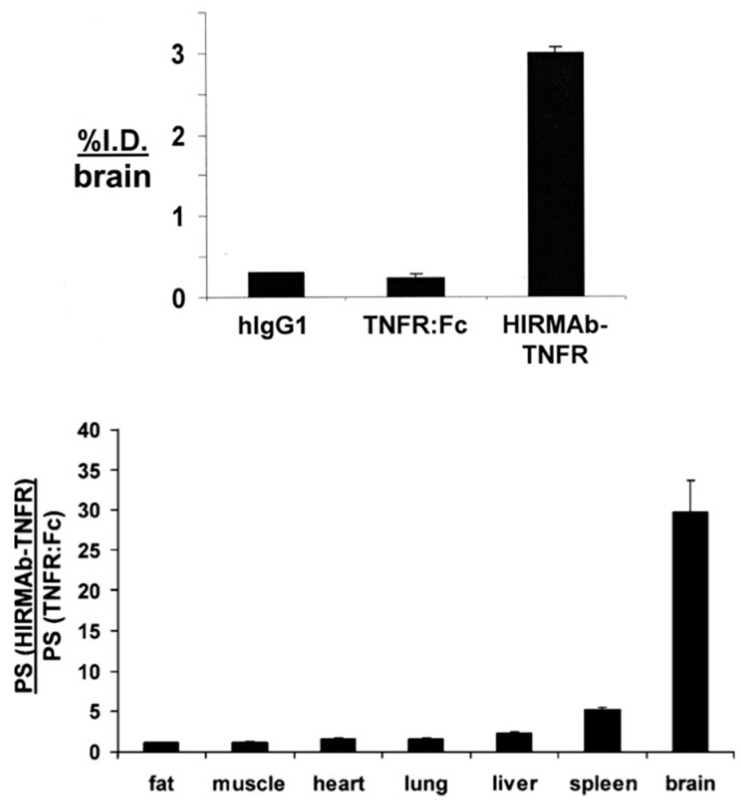
Selective targeting of a TNFR decoy receptor pharmaceutical to the primate brain as a receptor-specific IgG fusion protein. This fusion protein, HIRMAb-TNFR comprises a transport domain targeting the BBB human insulin receptor and the TNFR ECD as therapeutic domain. The brain uptake and peripheral biodistribution of the HIRMAb-TNFR were investigated in the rhesus monkey and compared with those of the control TNFR:Fc, etanercept, with [^125^I]-Bolton–Hunter reagent-labeled articles. The HIRMAb-TNFR fusion protein was transported across the BBB, producing a brain uptake of 3% ID. On the other hand, the non-brain-penetrating TNFR:Fc produced a brain uptake comparable to IgG1, which is confined to the blood compartment in the brain (**top**). The ratio for the organ permeability–surface area (PS) of the HIRMAb-TNFR relative to the organ PS for the TNFR:Fc in the rhesus monkey approximates 1 (**bottom**), as both molecules are transported into the peripheral organs. The PS ratio was >30 in the brain, as just the HIRMAb-TNFR is transported across the BBB and into the primate brain. From [30] with permission.

**Figure 10 pharmaceutics-14-01476-f010:**
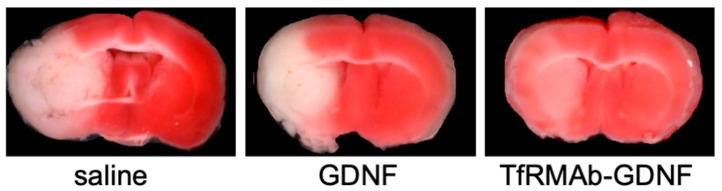
Neuroprotection of the mouse TfRMAb-GDNF fusion protein in a reversible middle cerebral artery occlusion (MCAO) stroke model. Brain coronal sections were obtained 24 h after MCAO and stained with 2,3,5-triphenyltetrazolium chloride (TTC), and representative brains are shown in the figure. Saline, 1 mg/kg BW mouse TfRMAb-GDNF fusion protein, and an equimolar dose of GDNF (0.17 mg/kg BW), were injected IV 45 min after MCAO. The mouse TfRMAb-GDNF fusion protein produced a 30% reduction in cortical stroke volume (**right**) compared with the control treated with saline (**left**), whereas GDNF alone had no effect on stroke volume (**center**). From [125] with permission.

**Table 1 pharmaceutics-14-01476-t001:** Brain-penetrating human IgG fusion proteins.

IgG Fusion Protein ^1^	Therapeutic Domain	Indication	Reference
HIRMAb-IDUA (valanafusp alpha)	Iduronidase (IDUA)	Hurler syndrome (MPS I)	[22]
HIRMAb-IDS	Iduronate-2-sulfatase (IDS)	Hunter syndrome (MPS II)	[23]
TfRMAb-IDS (pabinafusp alfa)	Iduronate-2-sulfatase (IDS)	Hunter syndrome (MPS II)	[24]
HIRMAb-ASA	Arylsulfatase A (ASA)	Metachromatic leukodystrophy *	[25]
HIRMAb-SGSH	Sulfamidase (SGSH)	Sanfilippo A (MPSIIIA) *	[26]
HIRMAb-NAGLU	N-acetyl-alpha-D-glucosaminidase (NAGLU)	Sanfilippo B (MPSIIIB) *	[27]
HIRMAb-ASM	Acid shingomyelinase (ASM)	Niemann–Pick A/B *	[28]
HIRMAb-HEXA	Hexoaminidase A (HEXA)	Tay–Sachs *	[28]
HIRMAb-PPT1	Palmitoyl-protein thioesterase (PPT1)	Batten Type 1 *	[28]
HIRMAb-GLB1	β-galactosidase (GLB1)	GM1-gangliosidosis *	[28]
HIRMAb-Aβ bispecific antibody	Anti-Aβ amyloid single-chain Fv antibody (scFv)	Alzheimer’s *	[29]
HIRMAb-TNFR	Tumor necrosis factor decoy receptor (TNFR)	Parkinson’s, ALS, Alzheimer’s, and/or stroke *	[30]
HIRMAb-EPO	Erythropoietin (EPO)	Parkinson’s, Alzheimer’s, and/or Friedreich ataxia *	[31]
HIRMAb-GDNF	Glial-cell-derived neurotrophic factor (GDNF))	Parkinson’s, stroke, and/or drug/EtOH addiction *	[32]
HIRMAb-BDNF	Brain-derived neurotrophic factor (BDNF)	Stroke, neural repair *	[33]
HIRMAb-Avidin	Any mono-biotinylated therapeutic	Various	[34]

^1^ The transport domain of these human fusion proteins is a monoclonal antibody directed to the human BBB insulin receptor (HIRMAb) or the transferrin receptor (TfRMAb). The therapeutic domain of the fusion protein and its application are listed for the corresponding IgG fusion protein. * Indication has a primary CNS disease burden.

**Table 2 pharmaceutics-14-01476-t002:** Brain-penetrating mouse IgG fusion proteins.

IgG Fusion Protein ^1^	Therapeutic Domain	Indication	Reference
TfRMAb-IDUA	Iduronidase (IDUA)	Hurler syndrome (MPS I)	[35]
TfRMAb-IDS	Iduronate-2-sulfatase (IDS)	Hunter syndrome (MPS II)	[36]
TfRMAb-SGSH	Sulfamidase (SGSH)	Sanfilippo A (MPSIIIA) *	[37]
TfRMAb-Aβ bispecific antibody	Anti-Aβ amyloid single-chain Fv antibody (scFv)	Alzheimer’s *	[38]
TfRMAb-TNFR	Tumor necrosis factor decoy receptor (TNFR)	Parkinson’s, Alzheimer’s, and/or stroke *	[39]
TfRMAb-EPO	Erythropoietin (EPO)	Parkinson’s, Alzheimer’s, and/or stroke *	[40]
TfRMAb-GDNF	Glial-cell-derived neurotrophic factor (GDNF))	Parkinson’s, and/or stroke *	[41]
TfRMAb-Avidin	Any mono-biotinylated therapeutic	Various	[42]

^1^ The transport domain of these mouse fusion proteins is a monoclonal antibody directed to the mouse BBB-transferrin receptor (TfRMAb). The therapeutic domain of the fusion protein and its experimental application are listed for the corresponding IgG fusion protein. * Indication has a primary CNS disease burden.

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
