# Peer review of "IgG Fusion Proteins for Brain Delivery of Biologics via Blood–Brain Barrier Receptor-Mediated Transport"

_pharmaceutics, 2022, doi:10.3390/pharmaceutics14071476_

Round 1

Reviewer 1 Report

The proposed review is of great interest to the field, well written, documented  and illustrated. The only corcern of the reviewer is the use of the terms 'conclusions and future perspectives' for section 9.  The reviewer considers that this section is presented as an overview of the whole manuscript. An overview is a very good idea to simplify the dissemination of the final message of the review.  A title for section 9 could be 'Overview and future prospects'. Therefore, in line 878, 'In summary' could be replaced by 'In conclusion'.

Author Response

The author is grateful to the reviewer's valuable comments that improved the manuscript. The recommendations of the reviewer have been incorporated in the revised version of the manuscript.

Reviewer 2 Report

This is a comprehensive review of antibody fusion proteins, which can be transported over the brain-blood barrier, utilizing “molecular Trojan horse” transport via transcytosis mediated either by HIR or transferrin receptor. This is an extremely timely topic, which will in the future even gain in the importance, regarding currently reported successes in the field, as well as the aging society, and is bound to make a difference to human health. This review describes the protein engineering strategies to derive the fusion proteins able to cross the BBB and exerting their therapeutic effect, their diverse mechanisms of action, the results of their application as well as their pharmacological properties (distribution, uptake into the brain, clearance, etc.) as well as, importantly, their side effects.

My comment would be that the chapter on molecular formats does not include the fusion with genetically engineered Fc domain, such as described in reference 50, which is a promising novel approach in the field, and advanced in clinical testing. A discussion on a possible effect of the smaller molecular size compared with the full-length antibody formats on the pharmacological properties would be an interesting addition, as well as an expansion of the discussion on “moderate” to high receptor affinity of the transferring moiety of the fusion protein, briefly addressed in the lines 151-156, and more details on discoveries on the importance of receptor binding valency. These are aspects which will critically influence the design of future therapeutics.

Another format that is promising in the BBB transfer are single domain antibodies (nanobodies, vNARS, etc.) but they feature other (mostly reporter) fusion proteins and are mainly used for imaging. In author’s view, could this format be explored for the delivery of biologics and which limitations would have to be overcome?

In the text, the sentences are sometimes repeated (please see comments below), and the use of several abbreviations makes the scientifically high-level text difficult to read. Certain Figures would profit of being redesigned for this publication (small fonts, low resolution of elements, additional points could be presented). Please find below a list of minor remarks which I hope you will find helpful.

Figure 2. Labels in the Figure are very small.

Lines 123-124: I would propose to list full names of the diseases: Batten disease Type 1, Tay-Sachs disease, Niemann-Pick disease Types A and B.

Table 1: please capitalize the therapeutic domains uniformly. Last row: typo, therapeutic

Line 191: Glucosaminoglycan, unless you mean glycosaminoglycans (also throughout the text)

Line 216: From

Line 220: six-months-old? The sentence is the same as in line 229: Legend to Figure 4?

Line 228: MPS-I, is that the same as MPSI?

Line 232: inclusion bodies

Lines 240 and 251: MPS I, is that the same as MPSI?

Line 244: was described

Line 266: Elaprase should be in capitals, as it is a commercial name

Line 283: 20 micrometer?

Line 283: 2h, not 2 Hrs

Line 284: scans were produced

Line 285: Bolton-Hunter reagent

Line 342, IP, intraperitoneal, the abbreviation is not defined previously in the text

Line 344: IV, intravenous, the abbreviation is not defined previously in the text

Line 346: SC, subcutaneous, the abbreviation is not defined previously in the text

Lines 389 and 390: sphingomyelinase                             

Line 425: neuronal ceroid lipofuscinosis

Line 470: knobs-into-holes

Lines 470-471: which produces (or delivers) antibodies composed of two heterologous half-antibody molecules

Line 493: linker :)

Line 493: tetravalent

Line 519: microliter

Lines 512-526: A, B, C: have different punctuation in the Figure Legend and in the Figure they are sometimes caps and sometimes not

Line 534: PSAPP: abbreviation not defined before

Line 537: C-terminus

Line 574: Bolton-Hunter reagent

Line 588: 6-hydroxydopamine

Line 630: 6-hydroxydopamine

Line 639: glial cell derived neurotrophic factor

Line 668: 6-hydroxydopamine

Line 791: for numbering, please provide the source and ID of the sequence you are referring to

Lines 803-804: would be better as a single sentence (sciatic nerve, overall suggesting…)

Line 838: bispecific

Author Response

The author is grateful to the reviewer's valuable comments that improved the manuscript. The manuscript has been revised as per the recommendations of the reviewer.

General comments

The reviewer mentioned that discussion on other brain drug delivery technologies, like engineered Fc domain fusion proteins, nanobodies and/or vNARS, would be an interesting addition to this review article. The author agrees with the reviewer; however, the author respectfully submit that the aim of the present article was to focus on the review of brain penetrating IgG-fusion proteins, and not on other potential technologies as those are reviewed in other articles of the special issue on Advanced Blood-Brain Barrier Drug Delivery, which includes the present work.

Specific Comments

  1. Figures 2 and 8 were revised as recommended by the reviewer.
  2. All typos identified by the reviewed were corrected in the revised version of the manuscript.

Reviewer 3 Report

In this review, the author provided an overview of re-engineering biotherapeutics to enable BBB transport and brain penetration for the treatment of brain and spinal cord disorders. A broad range of brain penetrating IgG-fusion proteins was discussed in the manuscript and demonstrated their efficacy through various animal models of CNS disorders. Overall, the review is well written and clearly structured, easily accessible with broad readership. The author has done a good job in bridging brain penetrating IgG-fusion proteins and their applications in neurological disorders.

Minor suggestions for improvement.

* It would be interesting to find out any possible challenges might be facing for the scaling up pharmaceutical manufacturing.

* More description about the use of IgG-fusion proteins in terms of their pharmacokinetics, immune response, and tissue toxicology. Perhaps summarise/compare in a table?

Author Response

The author is grateful to the reviewer's valuable comments that improved the manuscript. The recommendations of the reviewer have been incorporated in the revised version of the manuscript.

Specific comments

  1. The scaling up of the manufacturing process is now discussed in section 8 of the revised manuscript.

Enhanced description of the fusion proteins is now presented in section 8 of the revised manuscript.